

# A deep learning-based approach for emotional analysis of sports dance

Qunqun Sun[1] and Xiangjun Wu[2]

[1] Department of Physical Education, Qiannan Normal College for Nationalities, Dunyun, Guizhou, China
[2] College of Sports Science, Jishou University, Jishou, Hunan, China

## ABSTRACT

There is a phenomenon of attaching importance to technique and neglecting emotion in the training of sports dance (SP), which leads to the lack of integration between movement and emotion and seriously affects the training effect. Therefore, this article uses the Kinect 3D sensor to collect the video information of SP performers and obtains the pose estimation of SP performers by extracting the key feature points. The Arousal-Valence (AV) emotion model, based on the Fusion Neural Network model (FUSNN), is also combined with theoretical knowledge. It replaces long short term memory (LSTM) with gate recurrent unit (GRU), adds layer-normalization and layer-dropout, and reduces stack levels, and it is used to categorize SP performers' emotions. The experimental results show that the model proposed in this article can accurately detect the key points in the performance of SP performers' technical movements and has a high emotional recognition accuracy in the tasks of 4 categories and eight categories, reaching 72.3% and 47.8%, respectively. This study accurately detected the key points of SP performers in the presentation of technical movements and made a major contribution to the emotional recognition and relief of this group in the training process.

# INTRODUCTION

Sports dance (SP) is an artistic form that relies on the human body's movements to convey expression, making the human body a distinctive artistic medium. Despite its competitive nature, SP is often perceived as a technologically-driven performance art, overlooking the crucial role of emotion within the dance. Emotion serves as the primary element of dance art, as actions stem from highly stimulated emotions. Emotion is manifested through physical force, while the external movements are governed by internal emotions. The speed, intensity, and sequencing of actions are dictated by the underlying emotions. The language of emotions expressed through the body possesses elementary action primitives and their combinations, forming the basis for emotion computation based on body movements (*Fu et al., 2020*).

Analyzing the emotional aspects of the entire body poses challenges compared to traditional analysis that focuses on facial expressions. Limb movements in three-dimensional space encompass six degrees of freedom, allowing for a myriad of possible action combinations. Moreover, the difficulty in emotion recognition and classification

Corresponding author
Xiangjun Wu, wxj@jsu.edu.cn

lies in the variations among individual movements. Different individuals tend to express the same emotion in different ways, amplifying the challenges of classification (*Zhang & Zhang, 2022*). Scholars such as *Kang, Bai & Pan (2019)* have explored the interplay between limb and root joint constraints in individual and group motion scenes. *Vander Elst, Vuust & Kringelbach (2021)* delved into the description and correlation of dance movements, establishing emotional connections between music and dance in a shared space. Numerous studies have been conducted on emotional research in dance. However, some scholars focus solely on a single type of emotion or overlook the emotional aspects when analyzing the relationship between dance and other factors through computer analysis (*Wang & Chen, 2021*; *Wang, 2019*). These studies often concentrate on specific dance styles, limiting their scope of application. Additionally, some scholars have developed emotional models and explored their relevance to dance but rely on manual annotation of emotional types rather than implementing programming languages for emotional analysis within their systems (*Fang et al., 2018*).

Therefore, the video information of dancesport performers was collected by Kinect three-dimensional sensor. The pose estimation of dancesport performers was obtained by extracting key feature points, and FUSNN realized the emotional classification of dancesport performers.

## LITERATURE REVIEW

The advancement of deep learning has fostered the advancement of emotional recognition pertaining to human body posture, thereby facilitating the analysis of emotional attributes within the realm of SP. *Bianchi-Berthouze & Kleinsmith (2003)* provided a comprehensive depiction of posture by considering the angles and distances between various body joints. Subsequently, they devised an emotional pose recognition system that utilizes an associative neural network to map the pose feature set onto distinct emotion categories. Furthermore, *Bernhardt & Robinson (2007)* conducted an analysis of emotional expression in unadorned bodily movements through statistical measurements of kinematics. They employed 15 joints to represent the sampled skeletal structure and standardized the local body coordinate system based on body shape, thereby achieving invariant rotation and proportional measurements. By means of threshold analysis of limb energy, each action can be segmented, and the presence of limb movement within key frames can be discerned. The Laban movement analysis (LMA), widely employed in elucidating the principles of human dance, has recently provided inspiration for investigating the emotional facets of human posture. *Aristidou, Charalambous & Chrysanthou (2015a)* developed an array of 86-dimensional 3D body features derived from LMA theory, which facilitated the classification of emotions conveyed through bodily movements within dramatic performances. They defined parameters such as maximum/minimum/average distance, velocity, and acceleration between joint points, employing extreme random trees and SVM algorithms to carry out emotion classification tasks. In contrast to alternative approaches, this methodology (*Luo et al., 2019*) proves particularly well-suited for emotive scenarios. *Ajili, Mallem & Didier (2019)* introduced a human action description language

that incorporates an underlying grammatical structure to quantify diverse forms of human emotions.

The application of the common emotion classification algorithms is wildly used. *Aristidou, Charalambous & Chrysanthou (2015b)* used SVM and RF to classify performance data sets. Using human–computer interaction technology, *Ho et al. (2020)* combines two recurrent neural networks (RNN) and BERT to achieve accurate multi-mode emotion recognition. *Yadav et al. (2022)* used (RNN) to identify the user's activity data. However, the performance of RNNs when encountering long input data is not ideal. Limited by the problem of gradient disappearance, RNN has poor long-term memory ability. *Sapiński et al. (2019)* used LSTM to selectively memorize and forget input information by gate circuit, solving this problem.

This article uses non-performance data sets with high feature dimension and complex feature space and refer to the FUSNN model (*Zhang & Liu, 2019*). In the part of the deep macro network (MAC-NN), to simplify the model to ensure its performance of the model, BGRU is used instead of BLSTM. In addition, the dropout layer is added to the deep micro-network (MIC-NN) further to enhance the overfitting ability of the model.

## SP MOVEMENT RECOGNITION BASED ON A 3D SENSOR

### Kinect scene information extraction

Kinect bone tracking can track, at most, the data of two pairs of human bones, which is meaningful for SP competitions are performed mainly by two persons. Tracking can be divided into two modes: active tracking mode and passive tracking mode. In the active tracking mode, the user's skeleton data can be obtained using the key frame reading function. In the passive mode, the skeleton tracking of six people can be supported. Still, only the detailed bone data of two users can be obtained, and the other four people can only track and obtain the position information, but not the bone data. For example, when six people stand before Kinect, it can provide the location information and detailed joint point data of two (belonging to the active mode). In contrast, the other four (belonging to the passive mode) can only provide position information, and specific hand and head joint data cannot be provided. Therefore, this article adopts the active motion tracking mode of SP video.

Kinect sensor has three coordinate systems: screen, depth image, and bone space coordinate systems. The Kinect camera coordinate system is shown in Fig. 1. For the convenience of subsequent expression, this coordinate system is defined as skeleton space (coordinate). The coordinate origin is the centre of the infrared camera, the positive half-axis of the $X$ axis is the left-hand direction facing the Kinect viewing angle, the positive half-axis of the $Y$-axis is above the Kinect, and the $Z$-axis is on the optical axis of the infrared camera, which is perpendicular to the two-dimensional image plane.

Where $O_w$ is the world coordinate system, $O_c$ is the camera coordinate system, OUV is the image pixel coordinate system, and O is the physical image coordinate. The camera coordinate system is Kinect's coordinate system. When Kinect captures the content $w(x_w, y_w, z_w)$ in the world coordinate system, it needs to be converted to the camera

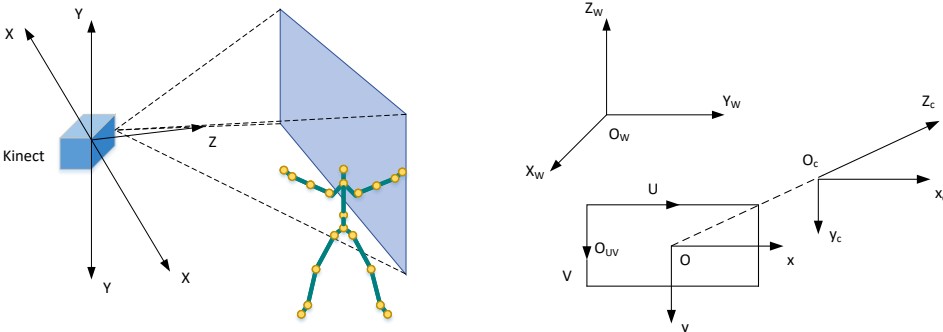

**Figure 1   Kinect sensor coordinate system diagram.**

coordinate system, and the conversion process is as follows:

$$\begin{bmatrix} x_c \\ y_c \\ z_c \\ 1 \end{bmatrix} = \begin{bmatrix} R_{3\times3} & T_{3\times1} \\ 0 & 1 \end{bmatrix} \cdot \begin{bmatrix} x_w \\ y_w \\ z_w \\ 1 \end{bmatrix}. \tag{1}$$

The transformation process between coordinate system OUV and coordinate system O is:

$$\begin{bmatrix} u \\ v \\ 1 \end{bmatrix} = \begin{bmatrix} \dfrac{1}{dx} & 0 & u_0 \\ 0 & \dfrac{1}{dy} & 0 \\ 0 & 0 & 1 \end{bmatrix} \begin{bmatrix} x_c \\ y_c \\ 1 \end{bmatrix}. \tag{2}$$

Where, $u_0$ is the projection of U in coordinate system OUV on x cycle in coordinate system O. At this point, the transformation process of $w(x_w, y_w, z_w)$ in the coordinate system, OUV is:

$$z_c \begin{bmatrix} u \\ v \\ 1 \end{bmatrix} = \begin{bmatrix} z_x & 0 & u_0 & 0 \\ 0 & z_y & v_0 & 0 \\ 0 & 0 & 1 & 0 \end{bmatrix} \begin{bmatrix} R_{3\times3} & T_{3\times1} \\ 0^T & 1 \end{bmatrix} \begin{bmatrix} x_w \\ y_w \\ z_w \\ 1 \end{bmatrix}. \tag{3}$$

In the process of the above-mentioned Kinect, the scene information of the SP trainer can be converted into image information in Kinect.

## Human pose estimation

The pose estimation of the SP trainer is the key content of judging the training action standard. The original image information generated by Kinect contains SP's trainer and training scene. To extract the target of the SP trainer, it is necessary to detect the target. This article uses the inter-frame difference method to extract the training target.

### Key point detection

After extracting the feature contour of SP trainers, detecting the joint human points is necessary to facilitate the pose-matching calculation. Human joint points are some points

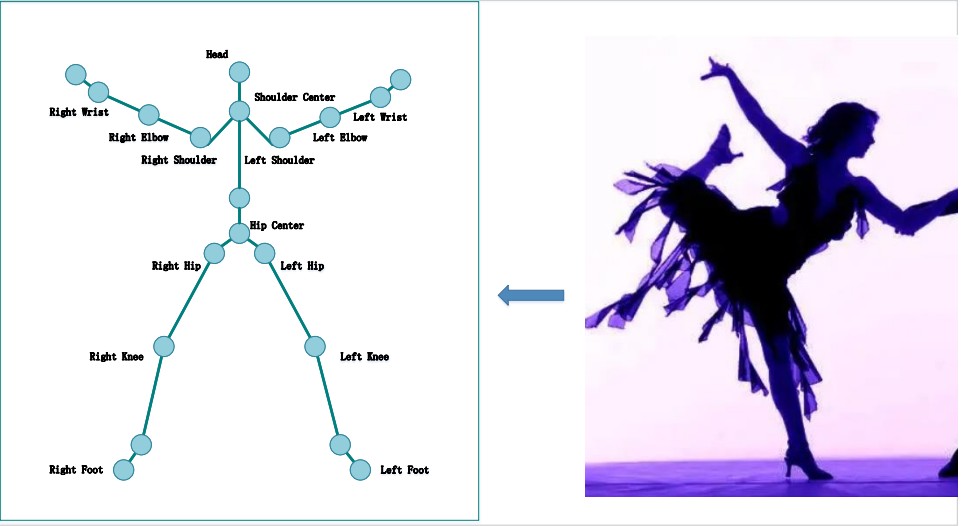

**Figure 2  Key points of SP.**

with position characteristics, generally composed of 15 to 20 points. According to the characteristics of SP training, the key points are selected, as shown in Fig. 2.

When detecting joint human points, first set a horizontal line at the top and bottom of the image, then move the top horizontal line down to the first intersection with the feature contour to judge the intersection point as the joint head point. After calculating the height of the trainer L, that is the difference between L1 and L2, the joint points of the human body can be detected according to the proportion of human joints. When the joint human points are detected, the detected joint point information can be matched with the joint point information of the sample library to complete the pose-matching calculation. This article uses the Euclidean distance to complete the matching calculation between the joint points. The process is as follows.

$$H = \sqrt{(x_1 - x_2)^2 + (y_1 - y_2)^2}. \tag{4}$$

Where $H$ is the matching value of attitude, and $(x_1, y_1)$ and $(x_2, y_2)$ are the coordinates of the detected node and the coordinates of the node in the sample database respectively.

Through the above steps, the human body poses estimation algorithm is realized to supervise the movements of the SP performers.

### Feature definition

The arm is the most emotional part of the body. The symmetry performance of the hand reflects the conscious tendency of posture, either approaching or avoiding a certain object. Many movements of the head and the trunk represent the mental movements of the head and the back of the body. Studies have shown that the energy index of body movement is an essential factor in distinguishing emotion (*Camurri, Lagerlöf & Volpe, 2003*). The primary forms of energy are velocity, acceleration and momentum. Therefore, each key point's velocity and acceleration scalar can be calculated by combining the 3D skeleton

information of two and three frames.

$$V(t) = \sqrt{v_x(t)^2 + v_y(t)^2 + v_z(t)^2} \tag{5}$$

$$A(t) = \sqrt{a_x(t)^2 + a_y(t)^2 + a_z(t)^2}. \tag{6}$$

Where, $v_i(t)$ and $a_i(t)$ represent the velocity and acceleration of each part of the body.

$$v_i(t) = \frac{i_{Coordinate}(t) - i_{Coordinate}(t-1)}{\Delta S}, i \in \{x, y, z\} \tag{7}$$

$$a_i(t) = \frac{v_i(t) - v_i(t-1)}{\Delta S}, i \in \{x, y, z\}. \tag{8}$$

# EMOTION CLASSIFICATION MODEL IN SP

## AV emotion model

Due to the fuzziness of emotion classification, this experiment uses the arousal-valence (AV) emotion model, as shown in Fig. 3.

From the above figure, we can intuitively see the distribution of different emotions and the correlation between different emotions. The closer the position is, the closer the emotion will be and the stronger the emotion will be.

## Emotion classification model based on BGRU

Referring to the FUSNN model, this article makes a series of modifications to the overfitting caused by the small number of non-performance data, high feature dimension and complex feature space. By training MAC-NN and MIC-NN, FUSNN captures the information of limb motion sequence in all aspects. In the MAC-NN part, to simplify the model on the basis of ensuring the performance of the model, BGRU is used instead of BLSTM. At the same time, a normalization layer is added between the stacked BGRU. Compared with the batch normalization layer, the normalization layer is more suitable for temporal neural networks, which normalizes different channels of the same sample and solves the problem that the sample statistical information can not reflect the global distribution when the later time unit is counted and is not affected by the size of batch size. Finally, the dropout layer is added to MIC-NN to enhance the overfitting ability of the model further.

The MIC-NN receives input consisting of the quantity of batch samples multiplied by the number of key frames, multiplied by the 3D feature data. The feature data for each frame is input in accordance with its temporal sequence, while the output corresponds to the classification of emotions. On the other hand, the MAC-NN receives input in the form of expanded data derived from 40 key frames, spanning 24 feature dimensions. This amounts to a 960-dimensional input data, with the resulting output aligning with emotion classification outcomes. The training process is that MIC-NN and MAC-NN are trained separately. Then the top-level feature vectors of the two networks are spliced together to input them into the FUSNN model for global joint training. Finally, the emotion classification results are output, as shown in Fig. 4.

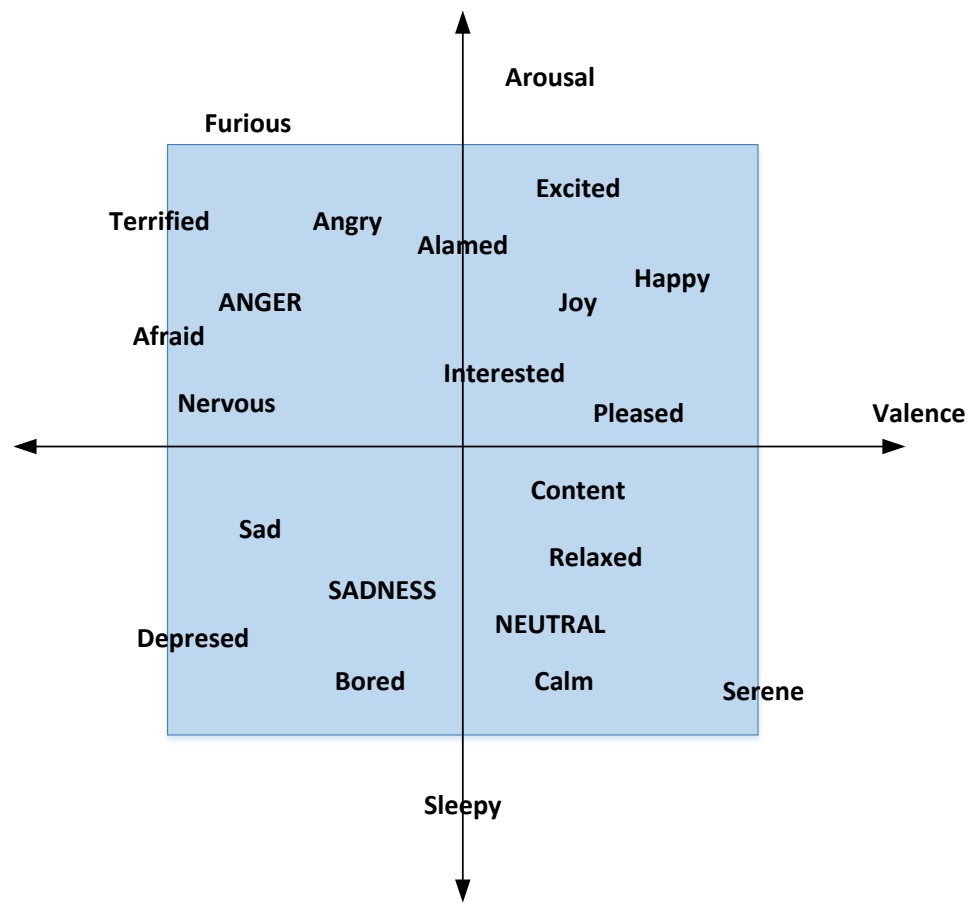

**Figure 3** AV emotion model.

# EXPERIMENT AND ANALYSIS

## Experimental environment

This article uses Matlab to build an experimental environment by combining Kinect 2.0 and Intel Corei7 processor on a computer with a 500 GB hard disk. Define the left shoulder, left elbow, left wrist, right hip, right knee and right foot as SP observation targets to test the detection effect of the proposed system.

## Data processing

The experimental data rate is two frames/s, and the total video frame length is 60 frames. Unlike the human eye's requirement for motion fluency, the model algorithm requires that the input data be rich in information and as simple as possible in structure. Redundant frame data will reduce the training convergence rate and bring more parameters to the model, resulting in overfitting and low robustness. Therefore, the trajectory curve simplification method is used. Based on the coordinate values of key points in frame data, the motion sequence is represented as a trajectory curve in 3D space, and the Lowe algorithm simplifies the curve. Suppose the horizontal error of the starting point and the

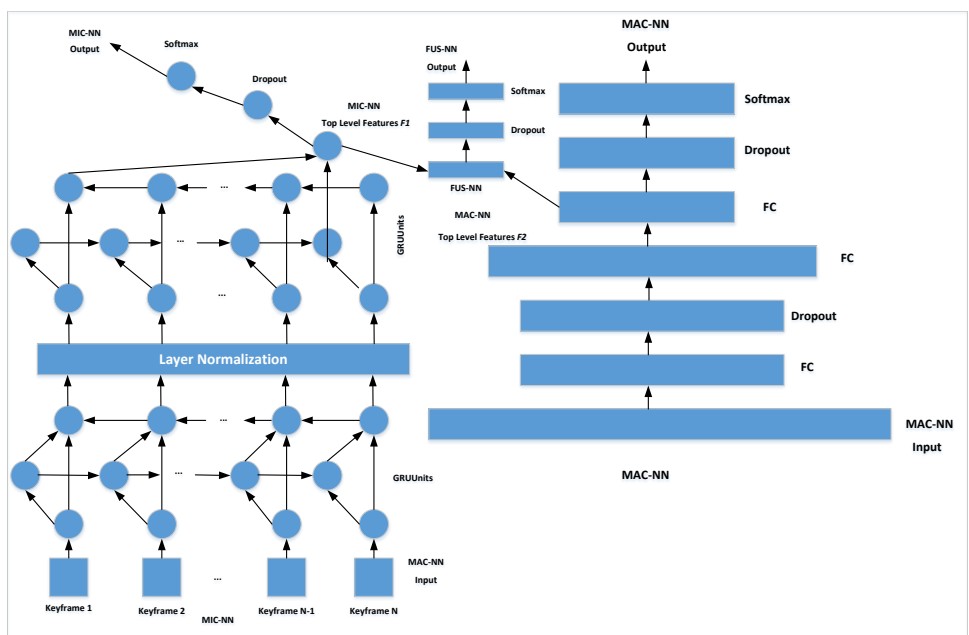

**Figure 4** **Structure of emotion classification model.**

curve's end point is larger than 2. In that case, it can be understood that if the deviation of the horizontal motion of the starting point and the curve's end point is greater than 2, the algorithm can be divided into two parts. Then the recursive operation is performed on the sub-lines until the error rate of each sub-line is small enough.

In this article, the point greater than the average significance of the action is defined as the candidate keyframe. As shown in Fig. 5, the number of keyframes of all experimental data samples under different error levels is calculated. The correlation curve between the number of key frames and the error level is drawn using the median as the representative value. After weighing the loss of data information and the need for input data compression, 40 is used as the key frame number of each input sample.

Considering the difficulty of experimental data collection and the problem that the sample data is small but the feature dimension is too high, PCA is used to reduce the feature dimension. Firstly, the maximum and minimum values of each column feature are normalized and compressed to [0,1]; Then, the covariance matrix is calculated for the original data, and a new dimension space is constructed by arranging the eigenvalues of each row of eigenvectors from large to small. The first 24 cumulative contribution rates, the component feature vectors whose cumulative eigenvalues account for more than 85%, are selected as the final feature space dimension.

## Results and Discussion
### Extraction accuracy of key points in SP
The test results of each key point are shown in Fig. 6. It can be seen that within 0 ∼100 frames, the motion track of athletes in the process of SP performance is well fitted with

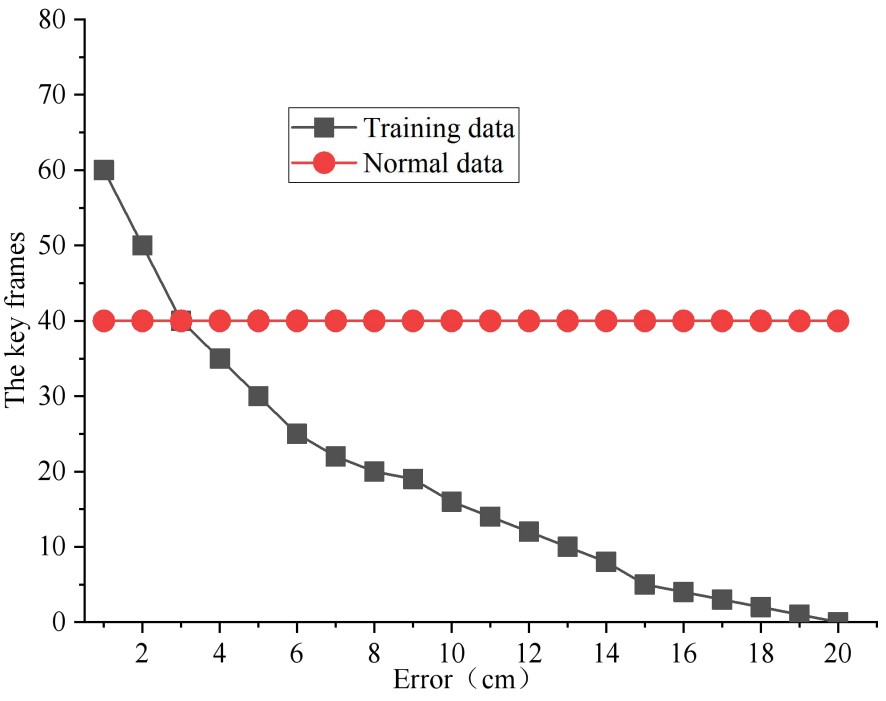

**Figure 5** **Key frame feature extraction.**

the sample database, but the error will be no more than 10°. In the performance process, the changing track of the key points of the performer is roughly consistent with the change of the position marked by the sample library. However, around 75 frames, the system detects that the trainer has a large action error. Therefore, the model proposed in this article can accurately detect the key points of SP performers in the performance of technical movements, thus laying the foundation for the follow-up study of emotion classification.

### Effectiveness comparison of different models

As shown in Fig. 7, the classification performance of the three traditional time series models under the above eight classification and four classification region division methods is compared, respectively. The performance of RNN is at the bottom because of its defect of easy gradient disappearance; BLSTM and BGRU can alleviate the loss caused by gradient vanishing through gate circuit operation and bidirectional propagation, but it still has the disadvantage of not making full use of long-term memory; FUSNN adds MAC-NN to process and classify the input information from the global point of view, and it also retains the advantages of LSTM for temporal data detail processing. Therefore, it has a better classification effect than the traditional time series model. In this article, BGRU-FUS-NN, designed based on non-performance action data sets, effectively optimizes over-fitting

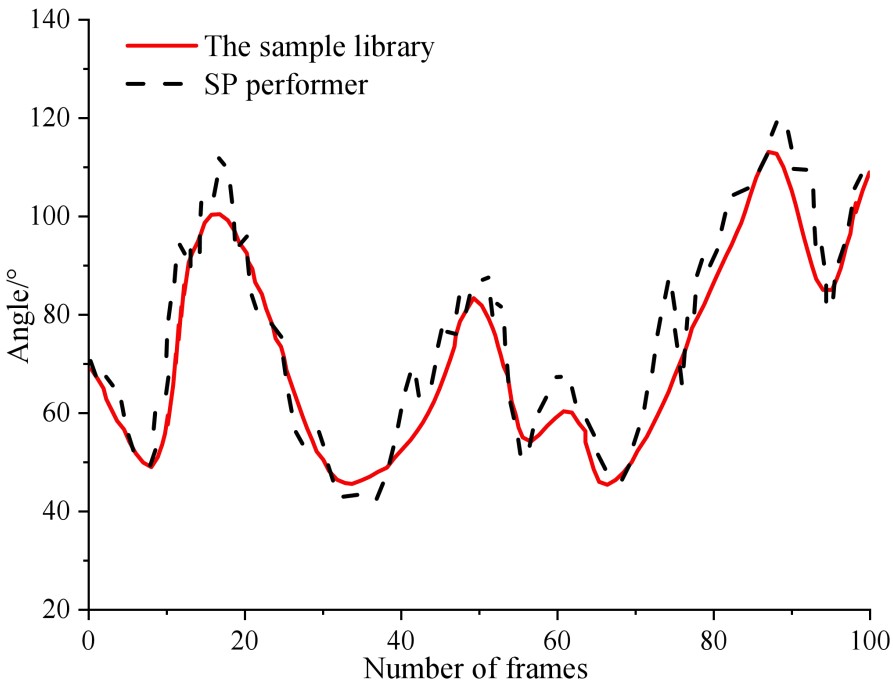

**Figure 6** Extraction results of key points in SP.

**Table 1** Comparative experimental results.

| Methods | Accuracy | F1 Score | AUC |
| --- | --- | --- | --- |
| RNN | 0.78 | 0.67 | 0.62 |
| LSTM | 0.84 | 0.78 | 0.71 |
| Bi-LSTM | 0.86 | 0.83 | 0.79 |
| CNN-Bi-LSTM | 0.94 | 0.91 | 0.87 |
| FUSNN | 0.98 | 0.95 | 0.92 |

problems by adding a layer-normalization layer and replacing LSTM with GRU and has the best average accuracy in both classification standards.

This article selected several models commonly used in emotion classification tasks for comparison, including the classical time series prediction model RNN, original LSTM, original Bi-LSTM, and CNN-Bi-LSTM combined with a convolutional neural network. The comparative experimental results are shown in Table 1. As can be seen from the data in the table, the experimental results of the proposed method are far superior to those of the classical models RNN and LSTM. Compared with CNN-Bi-LSTM combined with a convolutional neural network, the proposed method is also comprehensively ahead. In Accuracy, F1 Score and AUC, the proposed method is 0.04, 0.04 and 0.05 ahead of CNN-Bi-LSTM, respectively. These data illustrate the advantages of FUSNN proposed in this article.

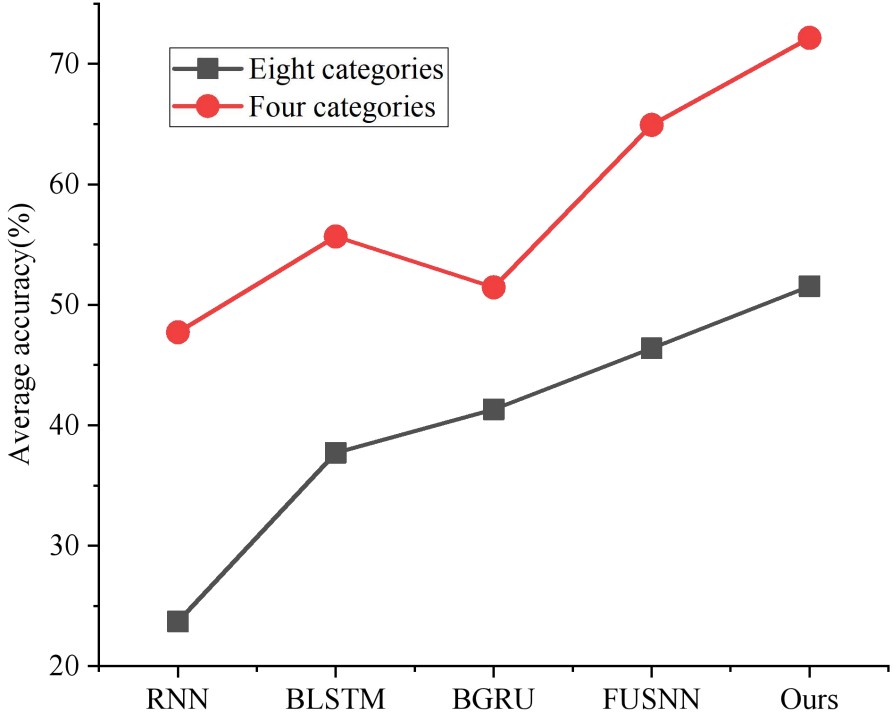

**Figure 7** **Classification accuracy of different models.**

### Emotional classification of SP movements

Figure 8 shows the result of the motion emotion confusion matrix of SP calculated by our model. The classification accuracy of excitement and tension was good; Their common ground is that they also belong to the high arousal area, and the subjects have a more significant difference in the body movements expressing intense emotions than others. On the contrary, the defined boundary between happiness and pleasure is vague, which is easy to confuse and complicated to distinguish; Sadness and other moderate arousal emotions also had a relatively low classification accuracy. However, sedation, with the lowest arousal, had a classification accuracy second only to tension and excitement.

The experimental results show that the emotion model can be divided into high, medium and low areas according to the arousal value. The model has a good classification effect in the high and low regions, respectively, corresponding to excitement, tension, and gentle emotion. In these two areas, the human body movement presents the same trend in the general direction, which can be mined from the characteristics of posture and energy. On the contrary, the correlation between the performers' actions and emotions is insignificant in the middle area, and the emotional postures vary from person to person. Hence, the classification effect is not ideal. It is worth mentioning that fatigue, a low arousal emotion, has a lower classification performance, contrary to this article's experimental results. The reason may be that performers are in a high-tension state in the competition process, which

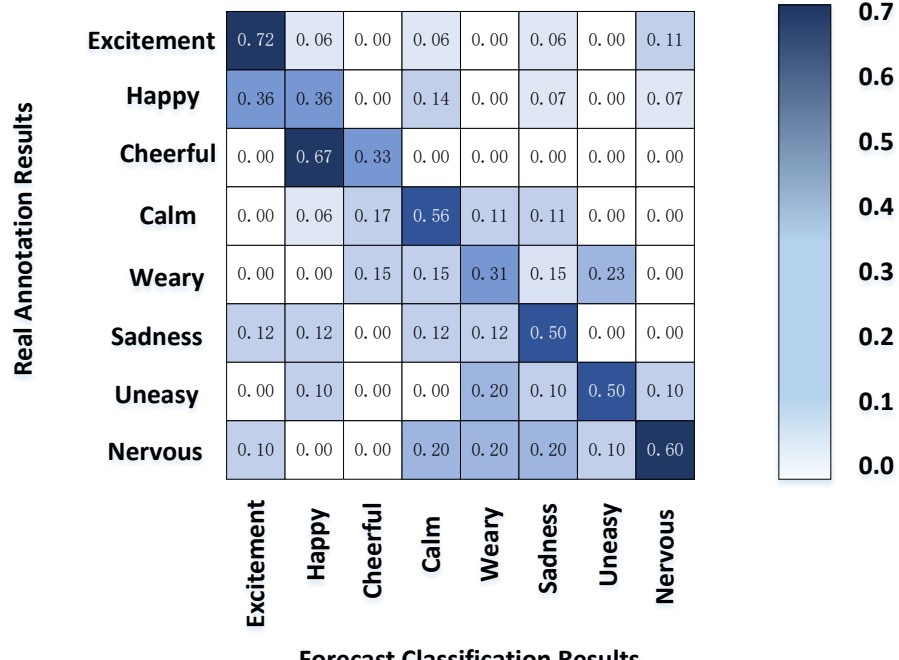

**Figure 8  Emotion classification of SP movements.**

is difficult to stimulate users' fatigue. Therefore, the number of samples is small, which leads to the classification results are not representative.

## CONCLUSION

SP is an expressive visual behaviour art in body movement skills. The outcome shows that the proposed model can accurately detect the key points of SP performers in technical action displays. Using BGRU to replace LSTM and other methods can optimize overfitting problems and have the best average accuracy under the two classification standards. In addition, the AV emotion model can be divided into high, medium and low regions according to the arousal value, which has a better classification effect in high and low regions, corresponding to excited, nervous and gentle emotions, respectively.

Emotion is the first element of dance art. The movement comes from people's highly excited feelings; emotion is expressed through force, and internal emotions govern external movements. This study is helpful to analyze better the relationship between the speed, weight and urgency of sports dance movements and emotions.

## ACKNOWLEDGEMENTS

I would like to thank the anonymous reviewers whose comments and suggestions helped improve this manuscript.

### Funding

This work was supported by the National Research Program for Philosophy and Social Sciences, the project number is 19BTY117 and 20BTY110. The funders had no role in study design, data collection and analysis, decision to publish, or preparation of the manuscript.

### Grant Disclosures

The following grant information was disclosed by the authors:
National Research Program for Philosophy and Social Sciences: 19BTY117, 20BTY110.

### Competing Interests

The authors declare there are no competing interests.

### Author Contributions

- Qunqun Sun conceived and designed the experiments, performed the computation work, authored or reviewed drafts of the article, and approved the final draft.
- Xiangjun Wu performed the experiments, analyzed the data, prepared figures and/or tables, and approved the final draft.

### Data Availability

The code is available in the Supplemental Files and at Zenodo: Wallace, Benedikte, Nymoen, Kristian, Martin, Charles P., & Tørresen, Jim. (2022). DeepDance: Motion capture data of improvised dance (2019) (1.0) [Data set]. Zenodo. https://doi.org/10.5281/zenodo.5838179

The AIST++ Dataset is available at: https://google.github.io/aistplusplus_dataset/factsfigures.html. The AIST Dance DB may only be used for academic research.

### Supplemental Information

Supplemental information for this article can be found online at http://dx.doi.org/10.7717/peerj-cs.1441#supplemental-information.

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
