# Peer review of "A deep learning-based approach for emotional analysis of sports dance"

_PeerJ Computer Science, doi:10.7717/peerj-cs.1441_

## Round 0.1 · original submission · Major Revisions

Although the study has some strengths, several areas need improvement. All reviewers have suggested revisions. One of the critical issues identified is the lack of explicit support and weak arguments to justify the proposed objectives, originality, and gaps that the study covers. The introduction should provide a more comprehensive explanation of these aspects to give the reader a better understanding of the study's significance. The author should also consider simplifying some of the sections in the paper, as they are overly detailed and difficult to follow. The language expression requires some modifications. Please see the detailed reviewers comments.

Reviewer 1 ·

Basic reporting

Overall, the paper seems ok and well-written. However, the writing could be further improved.

Experimental design

The experimental design is well explained, however, the results should be compared with counterpart papers for justification.

Validity of the findings

As I mentioned above, it is suggested to compare the results with any other relevant and recent work in order to compare the key findings.

Additional comments

I have the following concerns:

In this paper, Kinect three-dimensional sensor is used to collect the video information of dancesport performers, and the pose estimation of dancesport performers is obtained through the extraction of key feature points. In addition, based on the theoretical knowledge of the AV emotion model and fusion neural network model, GRU is used to replace (LSTM) to apply to the emotion classification of dancesport performers This is a valuable and interesting idea. However, with the current quality, this article cannot be published. Overall, the paper is good but I suggest incorporating the following changes to uplift the quality of the paper.

Please have someone competent in the English language and the subject matter of your paper go over the paper and correct it.
More data description of conclusions should be highlighted in the abstract;
The structure and content of this paper should be readjusted at the end of the introduction. This place is best expressed in another way.
The author should further explore the logical relationship between each title (including the subtitle) and the correlation with the corresponding content. For example, the title "Video information Conversion" in Section 3.1 cannot summarize the described content.

Authors should cite more internationally representative and influential articles;

The accuracy of numerical values is the concrete embodiment of whether the results of scientific experiments are rigorous.

All the data are calculated through the original data, so often not integers, which requires the author to unify the number of decimal points, most accurate to two decimal points can be.

The author should reflect on the research results, clarify the internal relations and development rules of things, and improve the further understanding of the research results from the aspects of depth and breadth, so as to help readers better understand and digest the research results.

Cite this review as

Reviewer 2 ·

Basic reporting

The development of deep learning promotes the emotion recognition of
human posture, which is helpful to analyze the emotional characteristics
in sports dance. The model proposed in this paper can accurately detect
the key points in the performance of technical movements of dancesport
performers. Using BGRU instead of LSTM and other methods, it can
effectively optimize the overfitting and other problems, and has the best
average accuracy under the two classification criteria. Although the
results of this paper have excellent value, there are also some problems,
some revisions needed to be revised to make sure that the manuscript can
be accepted. The commonly problems are as follows:
(1) There is insufficient support and weak arguments in support of the
objective that is proposed. In the final part of the introduction, the
proposed objectives, originality, and gaps that would be covered should
be better justified.
(2) Also a summary explaining how the author will perform the method
should be presented.
(3) The author should carefully check the notes and footnotes of the
formulas used in the paper, for example, Formula (4), and Formula (5) are
all lack of corresponding explanations.
(4) The author should delete some unnecessary expressions such as “and
record the position of the top horizontal line as L1. Similarly, when the bottom horizontal line is moved to the first intersection with the feature
contour, the intersection points are judged as left and right foot joint
points, and the position of the bottom horizontal line is recorded as L2.
(5) A lengthy description does not help the reader understand the model
used.
(6) The AV sentiment model mentioned in this paper is well applied in
the model, and the author sets its score range from −5 to +5, but it is not
reflected in the description of the results.
(7) The conclusion part needs to adjust the language expression, and the
elaboration of this part is too lengthy.
(8) The author needs to simplify this part. Meanwhile, the limitations of
the research and the future development direction need further
elaboration and analysis.
(9) There are also some problems in language expression in this paper,
which need to be modified. Please check the Chinese characters in the
replacement formula and the redundant space characters in the references.

Experimental design

The development of deep learning promotes the emotion recognition of
human posture, which is helpful to analyze the emotional characteristics
in sports dance. The model proposed in this paper can accurately detect
the key points in the performance of technical movements of dancesport
performers. Using BGRU instead of LSTM and other methods, it can
effectively optimize the overfitting and other problems, and has the best
average accuracy under the two classification criteria. Although the
results of this paper have excellent value, there are also some problems,
some revisions needed to be revised to make sure that the manuscript can
be accepted. The commonly problems are as follows:
(1) There is insufficient support and weak arguments in support of the
objective that is proposed. In the final part of the introduction, the
proposed objectives, originality, and gaps that would be covered should
be better justified.
(2) Also a summary explaining how the author will perform the method
should be presented.
(3) The author should carefully check the notes and footnotes of the
formulas used in the paper, for example, Formula (4), and Formula (5) are
all lack of corresponding explanations.
(4) The author should delete some unnecessary expressions such as “and
record the position of the top horizontal line as L1. Similarly, when the bottom horizontal line is moved to the first intersection with the feature
contour, the intersection points are judged as left and right foot joint
points, and the position of the bottom horizontal line is recorded as L2.
(5) A lengthy description does not help the reader understand the model
used.
(6) The AV sentiment model mentioned in this paper is well applied in
the model, and the author sets its score range from −5 to +5, but it is not
reflected in the description of the results.
(7) The conclusion part needs to adjust the language expression, and the
elaboration of this part is too lengthy.
(8) The author needs to simplify this part. Meanwhile, the limitations of
the research and the future development direction need further
elaboration and analysis.
(9) There are also some problems in language expression in this paper,
which need to be modified. Please check the Chinese characters in the
replacement formula and the redundant space characters in the references.

Validity of the findings

The development of deep learning promotes the emotion recognition of
human posture, which is helpful to analyze the emotional characteristics
in sports dance. The model proposed in this paper can accurately detect
the key points in the performance of technical movements of dancesport
performers. Using BGRU instead of LSTM and other methods, it can
effectively optimize the overfitting and other problems, and has the best
average accuracy under the two classification criteria. Although the
results of this paper have excellent value, there are also some problems,
some revisions needed to be revised to make sure that the manuscript can
be accepted. The commonly problems are as follows:
(1) There is insufficient support and weak arguments in support of the
objective that is proposed. In the final part of the introduction, the
proposed objectives, originality, and gaps that would be covered should
be better justified.
(2) Also a summary explaining how the author will perform the method
should be presented.
(3) The author should carefully check the notes and footnotes of the
formulas used in the paper, for example, Formula (4), and Formula (5) are
all lack of corresponding explanations.
(4) The author should delete some unnecessary expressions such as “and
record the position of the top horizontal line as L1. Similarly, when the bottom horizontal line is moved to the first intersection with the feature
contour, the intersection points are judged as left and right foot joint
points, and the position of the bottom horizontal line is recorded as L2.
(5) A lengthy description does not help the reader understand the model
used.
(6) The AV sentiment model mentioned in this paper is well applied in
the model, and the author sets its score range from −5 to +5, but it is not
reflected in the description of the results.
(7) The conclusion part needs to adjust the language expression, and the
elaboration of this part is too lengthy.
(8) The author needs to simplify this part. Meanwhile, the limitations of
the research and the future development direction need further
elaboration and analysis.
(9) There are also some problems in language expression in this paper,
which need to be modified. Please check the Chinese characters in the
replacement formula and the redundant space characters in the references.

Additional comments

The development of deep learning promotes the emotion recognition of
human posture, which is helpful to analyze the emotional characteristics
in sports dance. The model proposed in this paper can accurately detect
the key points in the performance of technical movements of dancesport
performers. Using BGRU instead of LSTM and other methods, it can
effectively optimize the overfitting and other problems, and has the best
average accuracy under the two classification criteria. Although the
results of this paper have excellent value, there are also some problems,
some revisions needed to be revised to make sure that the manuscript can
be accepted. The commonly problems are as follows:
(1) There is insufficient support and weak arguments in support of the
objective that is proposed. In the final part of the introduction, the
proposed objectives, originality, and gaps that would be covered should
be better justified.
(2) Also a summary explaining how the author will perform the method
should be presented.
(3) The author should carefully check the notes and footnotes of the
formulas used in the paper, for example, Formula (4), and Formula (5) are
all lack of corresponding explanations.
(4) The author should delete some unnecessary expressions such as “and
record the position of the top horizontal line as L1. Similarly, when the bottom horizontal line is moved to the first intersection with the feature
contour, the intersection points are judged as left and right foot joint
points, and the position of the bottom horizontal line is recorded as L2.
(5) A lengthy description does not help the reader understand the model
used.
(6) The AV sentiment model mentioned in this paper is well applied in
the model, and the author sets its score range from −5 to +5, but it is not
reflected in the description of the results.
(7) The conclusion part needs to adjust the language expression, and the
elaboration of this part is too lengthy.
(8) The author needs to simplify this part. Meanwhile, the limitations of
the research and the future development direction need further
elaboration and analysis.
(9) There are also some problems in language expression in this paper,
which need to be modified. Please check the Chinese characters in the
replacement formula and the redundant space characters in the references.

Cite this review as

Reviewer 3 ·

Basic reporting

The authors proposed an emotional analysis of sports dance based on deep learning framework. However, I have some concerns about the paper:

- In the abstract, the authors used the abbreviation SP with no pre-definition.

- The proposed method and contributions in the abstract should be rewritten.

- Section2 should be written as literature review , then recent studies should be added to identify different dimensions in emotion recognition.

- The proposed method is nor clear, the deep learning architecture and system should be explained.

- The experiments are very limited and did not conduct the idea.

Experimental design

- The experiments are very limited and did not conduct the idea.

Validity of the findings

No comments

Additional comments

No comments

Cite this review as

---

## Round 0.2 · accepted · Accept

I have assessed the revision myself, and the two reviewers are happy with the revisions.

Reviewer 1 ·

Basic reporting

Overall the paper is good shape now and improved in response to previous comments. Therefore I recommend it for publication

Experimental design

The current version of the paper has incorporated my suggestions therefore no more comments

Validity of the findings

I'm satisfied with the findings and novelty of this article

Additional comments

The paper is revised and I recommend it for publication

Cite this review as

Reviewer 2 ·

Basic reporting

In this article, the authors has proposed deep learning-based approach for emotional analysis of dance activities. This seems a different application of deep learning technique which may get wilder readership.

Experimental design

The revised version of the paper incorporated the design related comments and therefore this part seems to be good enough

Validity of the findings

Satisfied with the revised version and have no more comments

Additional comments

The overall comments seems to be addressed, therefore I have no more comments

Cite this review as